# Using 445 nm and 970 nm Lasers on Dental Implants—An In Vitro Study on Change in Temperature and Surface Alterations

**DOI:** 10.3390/ma12233934

**Published:** 2019-11-27

**Authors:** Sebastian Malmqvist, Anders Liljeborg, Talat Qadri, Gunnar Johannsen, Annsofi Johannsen

**Affiliations:** 1Division of Oral Diseases, Department of Dental Medicine, Karolinska Institutet, 141 52 Huddinge, Sweden; 2Division of Nanostructure Physics, Department of Applied Physics, KTH Royal Institute of Technology, 106 91 Stockholm, Sweden; 3Danakliniken Specialisttandvård, Praktikertjänst AB, 182 31 Danderyd, Sweden

**Keywords:** dental implants, diode lasers, temperature, scanning electron microscopy, in vitro

## Abstract

The aim of this study was to evaluate the safety of using a 445 nm laser on dental implants by comparing it with a laser with 970 nm wavelength. Two models, a pig mandible and glass ionomer cement, were used to evaluate the temperature increase in dental implants during laser irradiation with both wavelengths. Temperature was measured every second at four different places on the dental implants. Different power settings, effects of water cooling, distance of the laser fibre to the dental implant and continuous comparison to a pulsed laser beam were tested. Surface alterations on titanium discs after laser irradiation for 4 min at 2.0 W, were analysed in a scanning electron microscope (SEM). The maximum temperature and time to reach each of the thresholds were comparable between the 445 nm and 970 nm lasers. Neither the 445 nm nor the 970 nm wavelength showed any signs of surface alterations on the titanium discs. Using a 445 nm laser on dental implants is as safe as using a 970 nm laser, in terms of temperature increase and surface alterations. Applying a generous amount of cooling water and irradiating in short intervals is important when using lasers on dental implants.

## 1. Introduction

One of the great challenges in periodontology today and in the future is peri-implantitis, which is an inflammatory disease characterised by loss of the supporting peri-implant tissues [1]. Replacing missing teeth with dental implants has become a popular treatment option, with an estimated 12–18 million implants sold annually worldwide, and, of those, roughly 5.5–6 million are sold in Europe [2]. The prevalence of peri-implantitis at patient level has been reported to be approximately 22% (CI: 14–30%) in a meta-analysis [3], consisting of 11 studies from between 2005 and 2013. In a recent cross-sectional study [4] in a Spanish population, the prevalence was 24% (CI: 19–29%), which is in line with the previously mentioned meta-analysis [3]. Together these approximations imply that worldwide every year, 2.6–4.0 million patients are at risk of developing the disease [2,3].

The current treatment options of peri-implantitis are few and consist mainly of submucosal cleaning around the implants and removal of the inflamed soft tissue, and optimizing oral hygiene conditions [5]. Advanced lesions often require surgical treatment, which is both time-consuming and expensive for the patient [6]. Other treatment options that have been tested include adjunctive antibiotics and treatment with surgical lasers. Surgical lasers are a category of lasers with different wavelengths, which absorb well in certain tissues and are used with power settings high enough to remove those respective tissues [7]. The laser medium in each laser device determines the wavelength of the emitted photons, which, in turn, defines the absorption in the tissue. The operating parameters of the laser (e.g., wavelength, power setting, continuous or pulsed mode), the duration of treatment, the type of fibre, and its distance from the treatment site, determine its effect on tissue. Since there is limited evidence as to which treatment procedures of peri-implantitis are most effective [6], the importance of testing new treatment methods is obvious, not only for their clinical effect, but also to reduce the suffering of, and costs to, the patient.

In a widely cited study by Eriksson and Albrektsson [8], the reaction of the bone to temperature increases of different magnitudes was investigated. Rabbit tibia seems to be negatively affected if the temperature increases above 47 °C for 1 min, which corresponds to an increase of 10 °C. This study focuses on bone growth in a canal in a titanium implant, but not the effect of temperature increase on osseointegration of a titanium implant. Trisi et al. [9], on the other hand, examined the osseointegration in sheep after heating the osteotomic site to 50 °C or 60 °C for one minute before placement of the implant. They found no implant failures after 2 months of healing, but the 60 °C showed histological signs of inflammation and significant crestal bone resorption. It is difficult to deduce a threshold for human bone based on these studies, but one should certainly strive to minimise the time with a temperature above +10 °C.

Previous studies done in vitro on temperature increases in and around dental implants have shown great discrepancies between different wavelengths of lasers [10,11]. Also, studies on alterations of the dental implant surface show conflicting results as to which wavelength of lasers should be used with limited risk for damaging the dental implant [12,13,14]. To our knowledge, no studies have yet looked at the potential deleterious effects of the 445 nm wavelength when directly irradiating a dental implant.

Our hypothesis is that a laser with a wavelength of 445 nm is safe to use on and around dental implants in terms of temperature increase and surface alterations. The aim of this study was to evaluate the safety of using a 445 nm laser on dental implants by comparing it with a laser with a 970 nm wavelength.

## 2. Materials and Methods 

### 2.1. Laser Device

The laser device used in this study was a SiroLaser Blue (Dentsply Sirona, Bensheim, Germany) which has incorporated three wavelengths—445 (± 5) nm, 660 (± 5) nm and 970 (-10/+15) nm. The primary wavelengths that were tested were 445 nm and 970 nm, while the 660 nm wavelength was used as an aiming beam with 1 mW power. The actual power output was measured seven times during the study, with an 841-PE Handheld Power Meter (Newport Corporation, Irvine, CA, USA), for both wavelengths, at 1.0 W and 2.0 W displayed output, respectively (Table 1). The diameter of the laser fibre was 320 μm. Distances and power settings varied with the different tests.

Temperature was measured with NiCr-Ni (type K) thermocouples (FTA3901, Ahlborn Mess- und Regelungstechnik, Holzkirchen, Germany), connected to a multimeter (Almemo 2690-8A, Ahlborn Mess- und Regelungstechnik, Holzkirchen, Germany). 

Two models were deployed to test the temperature increase and the tests were repeated three times unless stated otherwise. The first model was a dental implant placed in a glass ionomer cement (GIC) block, which is more convenient for repeated tests. The second model was a dental implant placed in a pig mandible (PM) to simulate the clinical setting. GIC was chosen for the majority of the temperature tests because it has a similar thermal conductivity to human bone. The thermal conductivity has been reported as 0.30–0.64 W/mK for GIC [15], depending on the density and composition of the material, 0.32 W/mK for pig shoulder bone [16], 0.68 W/mK for human cortical bone and 0.42 W/mK for human bone marrow [17]. The PM model, being more complex with soft tissue and mandibular bone composition similar to a human’s, is closer to the clinical situation in humans, but, due to the tissue decay at room temperature, it was used for few selected tests after the GIC tests.

Five thermocouples were placed on each model (Figure 1 and Figure 2); next to the irradiation site, at half of the distance to apex from the irradiation site, at the most apical part of the implant, inside the dental implant and, lastly, one measuring the room temperature approximately 0.5 m from the experiment. The temperature was registered every second starting a few seconds before the start of the tests and continuing at least 2 min after the laser was turned off for each respective test. Both thresholds of +10 °C and +20 °C were considered when analysing the temperatures.

#### 2.1.1. Glass Ionomer Cement Model

A block was made of glass ionomer cement (GC Fuji IX GP, GC Corporation, Tokyo, Japan) with dimensions approximately 28 × 20 × 13 mm. An Astra Tech OsseoSpeed EV 3.6 × 11 mm dental implant was placed in the GIC block (Figure 1). Holes were drilled in the block for thermocouples midway down and for the apical part of the implant.

The first set of tests in this model were different power settings (0.5 W, 1.0 W, 1.5 W and 2.0 W) for both wavelengths. The implant was radiated for 2 min on continuous wave (cw) mode. The laser handpiece was mounted on a stand and the optical fibre aimed 90 degrees towards the dental implant surface at 1 mm distance. In this configuration, all the light emitted at the end of the fiber hits the surface of the implant. Normally, when using the laser around dental implants in patients, the angle would be between 45 degrees to parallel to the surface. This results in a part of the laser light not hitting the implant. Thus, our model represents a worst-case scenario for temperature increase in the dental implants. These tests were done to characterise the model and get a magnitude of the temperature increase.

The second set of tests aimed towards exploring different factors affecting the temperature increase. Pulsed mode for the 445 nm wavelength was compared to cw mode. Settings used were 3.0 W displayed power, a frequency of 10 Hz and a duty cycle of 17%, leading to an average power output of 0.51 W, which was compared to the cw mode of 0.5 W from the first set of tests. Different distances were also explored, from contact with the optical fibre tip and each whole millimetre up to 3 mm distance. Moving the handpiece manually along approximately half of the dental implant’s circumference, similar to what one could reach buccally or lingually in a patient setting, were also tested for the 445 nm wavelength, 1 W in cw mode, and for 2 min.

The third set tested different ways of cooling the implant with room temperature water in a syringe. Both wavelengths, 1 mm distance of fibre tip, 2 min of irradiation in total and cw mode, were used in every water test scenario. Manually continuously applying 20 mL water on the site during irradiation was tested for 1.0 W and 2.0 W. Then, different intervals of applying different amounts of water at 1.0 W were tested. The first test involved 30 s of irradiation, followed by the application of 5 mL of water, which was repeated three times to reach a total irradiation time of 2 min. Then next tests involved 30 s of irradiation with 2.5 mL of water. Lastly, 15 s of irradiation was followed by the application of 2.5 mL of water.

#### 2.1.2. Pig Mandible Model

An Astra Tech OsseoSpeed EV 3.6 × 11 mm dental implant was placed in an edentulous area in a whole pig mandible, with the gingival soft tissue still attached to the mandible bone (Figure 2). The drilling was performed in accordance with the manufacturers standard protocol and the final position of the neck of the implant was 2 mm above the bone level. The tests in the PM model were done after the GIC tests. The PM was kept frozen prior to the tests and was thawed in a bucket filled with room temperature water for 2 hours. After the thawing, the implant was irradiated for 2 × 2 min and the temperature was then allowed to settle again to balance around room temperature. This was to make sure that the PM was not locally cooler at the implant site due to incomplete thawing. The tests were not initiated until all four measuring sites reached a stable value at room temperature, likewise, the temperature had to stabilise between each test.

In the first set of tests, the handpiece was mounted statically. A power setting of 1.0 W in cw mode was used for both wavelengths, to allow comparison between the GIC and PM models. The last set of tests were clinical simulations, where the handpiece was moved manually back and forth during irradiation for 30 s, 5 mL of cooling water was applied, and then another 30 s of irradiation was followed by water cooling, continuing until a total of 2 min of irradiation.

### 2.2. Surface Alteration Tests

For the surface alteration tests, titanium discs with flat surfaces were analysed with a scanning electron microscope (SEM). The SEM pictures were taken with a FEI Nova 200 Dual Beam (combined SEM and Focused Ion Beam), located at Albanova NanoLab KTH, with an accelerating voltage of 10.0 kV, a probe current of 0.13 nA and focus distance of 5.0 mm.

Machined titanium discs and sandblasted and acid-etched titanium discs, with dimensions 10 mm diameter and 2 mm thickness, were used. To easier identify the irradiated area in the SEM, the discs were marked with a cross by a pen and, in one of the quadrants, a smaller cross was scratched into the metal with a sharp instrument (Figure 3). The discs were cleaned in an ultrasonic bath with isopropanol for 5 min, to remove dirt and fat from the discs so they did not interfere with the vacuum in the SEM. SEM pictures were then taken at the site marked for laser irradiation with different magnifications ranging from 75× (500 µm scale bar) to 25000× (1 µm scale bar). The discs were then removed from the SEM and irradiated with the 445 nm wavelength for 4 min at 2.0 W cw mode at a distance of 1 mm. After irradiation, the discs were placed back in the SEM and pictures were taken again at different magnifications.

### 2.3. Statistics

Due to the explorative nature of this study, descriptive statistics was used to analyse the results. Microsoft Excel 2016 (Microsoft Corporation, Redmond, WA, USA) was used to facilitate the analysis.

## 3. Results

### 3.1. Pig Mandible Model

In the static test, the time to reach the thresholds of +10 °C and +20 °C were similar for both the 445 nm and the 970 nm lasers (Table 2). The apical part of the implant reached the +20 °C mark quicker for the 445 nm laser compared to the 970 nm. Average temperatures were slightly higher for the 445 nm laser at three of the four thermocouples, midway down the implant, at its apical part and next to the irradiation site (Table 3).

In the clinical simulation test with 1.0 W power, the 445 nm and 970 nm lasers resulted in a similar temperature increase (Figure 4). The time it took to reach the +10 °C threshold at the midway thermocouple was 14 s and 15 s for the 445 nm and 970 nm lasers, respectively. The temperature did not reach +20 °C for either wavelength of laser at the midway thermocouple. The temperature did not completely return to its starting value when cooling with 5 mL for 8–10 s. However, the sites inside, midway and next to irradiation did not increase in temperature gradually with each irradiation interval. The temperature in the apical part, on the other hand, did gradually increase with each repeat.

### 3.2. Glass Ionomer Cement Model

The maximum change in temperature at all four of the measured sites was higher in the GIC model than in the PM model for both wavelengths (Table 4). Time to reach the +10 °C and +20 °C thresholds were shorter inside the implant and on the surface next to the irradiation site for the GIC model, but longer for the midway and apical sites.

Using the 445 nm laser with a pulsed beam, with comparable average power to continuous wave, increased the temperature inside, midway and next to the irradiation site of the implant (Table 5). At the apical part of the implant, there were miniscule differences between the pulsed and continuous modes. The higher peak power with the pulsed mode lead to a locally higher temperature than continuous mode.

The temperature decreased with increases in the distance of the laser’s fibre tip from the implant at the inside, midway and apical parts (Table 6). At a distance of 3 mm, the thermocouple next to the irradiation site was radiated directly, due to the spread of the beam at that distance. Moving the handpiece during irradiation generally led to a 10.2 to 4.6 °C lower temperature increase inside the implant compared to a static mounted handpiece. The midway and apical parts showed slightly higher temperature increases and the temperature increase was more uneven when moving the handpiece at these sites.

All the water tests had larger variations in temperatures during repeats than the dry tests. Continually applying water during the irradiation of the implant resulted in a lower average temperature for the 445 nm wavelength than 970 nm (Table 7). For the 1 W tests and the midway and apical sites, the +10 °C threshold was reached only for the apical site using the 970 nm laser.

There is a meaningful difference between using 2.5 mL and 5.0 mL cooling water after 30 s interval of laser irradiation, regardless of wavelength (Table 8). Using 5.0 mL of cooling water was therefore chosen for the PM tests. When using an interval of 15 s with 2.5 mL of water, the temperature gradually increased for every interval at the midway and apical thermocouples. The difference in average maximum temperature for the 445 nm wavelength, between the first and last interval was 6.57 (± 0.55) °C and 5.23 (± 0.23) °C at the midway and apical sites. This gradual increase was not seen for 30 s interval with 2.5 mL water. In this test, the drops in temperature between intervals were shallower than when using 5 mL of water.

### 3.3. Surface Alterations

There were no signs of surface alterations on the titanium discs, regardless of surface type, after laser irradiation for 4 min at 2.0 W power setting, and in continuous wave mode with the 445 nm wavelength (Figure 5 and Figure 6).

## 4. Discussion

In general, the differences between the two wavelengths were small in all the tests. This should translate to a slight advantage for the 445 nm wavelength when used clinically on patients. The reasoning for this is that the 445 nm laser has been shown to more efficiently cut soft tissue at a lower power setting than the 970 nm laser [18]. Working at a lower power setting results in a lower temperature increase, as seen in this present study. In terms of temperature increase when using different lasers on dental implant, one should use them during short intervals with generous water-cooling in between. This recommendation should be fairly easy to incorporate in the clinical use of the laser, due to the need to remove seared soft tissue and coagulated blood from the hot fibre tip when using it in a peri-implant or a periodontal pocket [19]. 

Valente et al. [10] reached lower temperatures for 810 nm, 980 nm and 1064 nm at both the apical and coronal parts of the implant. They used a similar sized implant (4 × 11 mm) and placed it in a small bone block. In the present study, we used a whole pig mandible with the gingival tissue still remaining and reached a temperature increase of 17.2 °C at the apical part for the 970 nm wavelength, whereas Valente et al. reported 13.7 °C for 980 nm and the same power setting. The difference could be partly explained by Valente et al. having the fibre tip at a distance of 3 mm, which we noticed in the present study affected the temperature. Valente et al. also noted a difference in temperature between the different wavelengths when using the same power settings (1.0 W in continuous mode). The change in temperature at coronal and apical, respectively, was, for 810 nm, 10.8 °C and 7.9 °C, for 980 nm 19.4 °C and 13.7 °C, and, lastly, for the 1064 nm, 23.6 °C and 5.2 °C. A difference between 810 m and 980 nm has also been observed by Leja et al. [11], showing temperature increases of 12.0 °C and 4.3 °C for the 810 nm, and, for the 980 nm, 31.5 °C and 8.6 °C. This is in contrast with the findings in the present study, which showed insignificant temperature differences between the 445 nm (39.7 °C and 18.0 °C) and 970 nm (37.0 °C and 17.2 °C) wavelengths in the PM model. Neither Valente et al. [10] nor Leja et al. [11] discuss possible reasons for the variation between wavelengths in their studies, even though their differences should be of interest when translating the tests to a clinical setting. Their measured temperatures vary by approximately 10 °C, which, in context, is a large difference, given that the threshold for hazardous effects on bone has been reported to be an increase of 10 °C [8].

The inside of, and areas next to, sites showed a higher maximum temperature for GIC than PM, which could be explained by the fact that the heat generated spreads to the surrounding area better in the pig model. A likely explanation for this is the difference in volume between the models, and that the PM had soft tissue. Soft tissue, saliva and a blood flow through living tissue could further decrease the temperature in vivo. Stein et al. [20] reported that blood flow in teeth decreases the temperature generated when debonding orthodontic brackets with the 445 nm laser. These cooling factors should put these in vitro results on the safe side when it comes to applying this present study to the clinical setting.

In the present study, no surface alterations were found for the 445 nm wavelength when radiating for 4 min at 2.0 W with a distance of 1 mm to the discs. Surface alterations on titanium discs and dental implants have been reported in other studies using different wavelengths, distances to the object, and power settings [12,13,14,21]. Romanos et al. [12] showed that using Nd:YAG (1064 nm wavelength) at 2.0 W and 4.0 W in contact resulted in, as they describe it, melting and damage to the surface of the discs. They did not see any surface alterations when using a diode laser (980 nm) with 5.0, 10.0 or 15.0 W continuous mode with a contact handpiece. Lee et al. [21] noted surface alterations with Er:YAG (2940 nm) 1.4 and 1.8 W (140 mJ/pulse and 180 mJ/pulse) in contact for 1 min, but not with 1.0 W. Stubinger et al. [13] saw the same with Er:YAG at 3.0 W and 5.0 W (300 mJ/pulse and 500 mJ/pulse), used for 10 s in contact with the disc. At 1.0 W, the Er:YAG did not result in surface alterations as well as when using CO_2_ laser (10600 nm) at 2.0, 4.0 or 6.0 W or diode laser (810 nm) at 1.0 or 3.0 W; both CO_2_ and diode lasers were used at a distance. It seems unreasonable that these findings of surface alterations are due to the wavelengths of the lasers, despite the authors’ claims. For example, that Nd:YAG laser used at 2.0 W resulted in melting, while a diode laser used at 15.0 W at a comparable wavelength did not, both with the same irradiation time and in continuous wave mode, seems unlikely. A more likely explanation is that the surface alterations are caused by using the lasers’ fibres in contact with the surfaces, which Giannelli et al. [14] have shown to cause scratches of the surface and not melting.

More studies are needed that compare the different wavelengths of lasers. These studies should use the same parameters (e.g., same average power) for the different lasers, to get comparable results of the temperature increases. There is a great variation in models used to in vitro test temperature development in dental implants. Studies looking at temperature change in the coronal part of dental implants already installed in humans should be possible, if one uses power settings on the safe side. Also, several studies have looked into surface decontamination of dental implants using Er:YAG (2940 nm) [22,23] or diode laser (810 nm) [24] in humans with peri-implantitis. Schwarz et al. [22] used Er:YAG with 1.0 W (100mJ/pulse, 10Hz) with water irrigation and Renvert et al. [23] used the same power settings but did not state that they used any water irrigation. Neither of these studies reported any deleterious effects, but they also did not report how long each implant was irradiated for. Bach et al. [24] used the 810 nm laser at 1.0 W for a maximum of 20 s intervals and reported no deleterious thermal effects. It should, therefore, be ethically viable to perform a clinical study measuring the temperature with a thermocouple at the coronal part of the dental implant in humans when using 1.0 W and 20 s maximum intervals. Since the 445 nm and 970 nm wavelengths yielded similar temperature increases, then it should be feasible that the protocol for the 810 nm laser could be applied to either of those wavelengths as well.

## 5. Conclusions

Within the limits of this study, using a 445 nm laser on dental implants is as safe as using a 970 nm laser in terms of temperature increase and did not result in any surface alterations. The 445 nm laser can thus be used for peri-implantitis treatment with limited risk of hazardous effects. 

## Figures and Tables

**Figure 1 materials-12-03934-f001:**
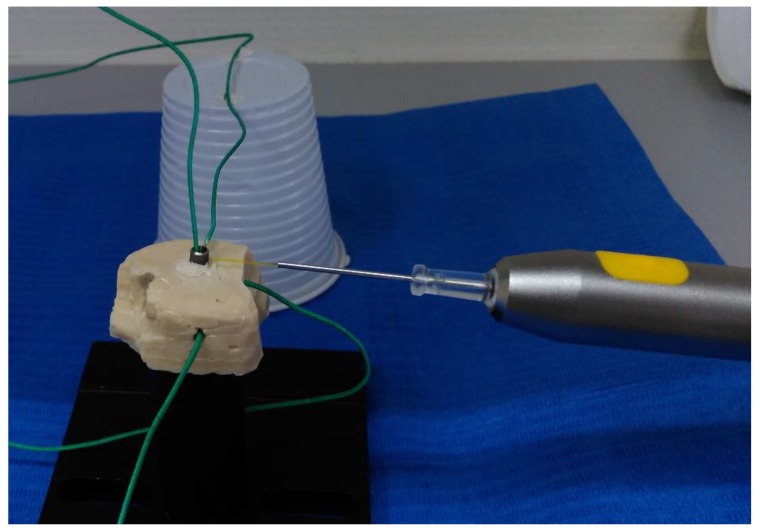
Glass ionomer cement block with attached thermocouples and laser handpiece mounted on a stand.

**Figure 2 materials-12-03934-f002:**
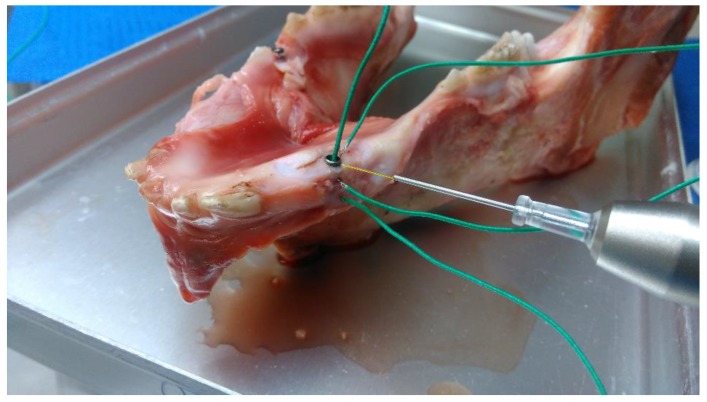
Pig mandible with dental implant showing the set up for the static temperature tests.

**Figure 3 materials-12-03934-f003:**
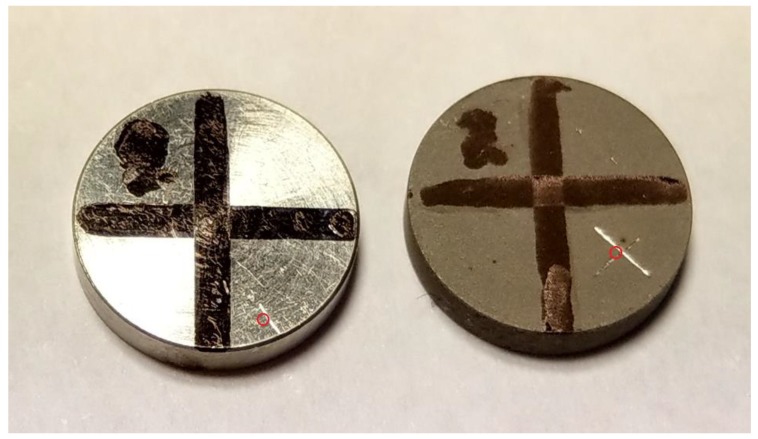
Titanium discs used, the site of irradiation marked with a red circle; machined surface to the left and sandblasted and acid-etched to the right.

**Figure 4 materials-12-03934-f004:**
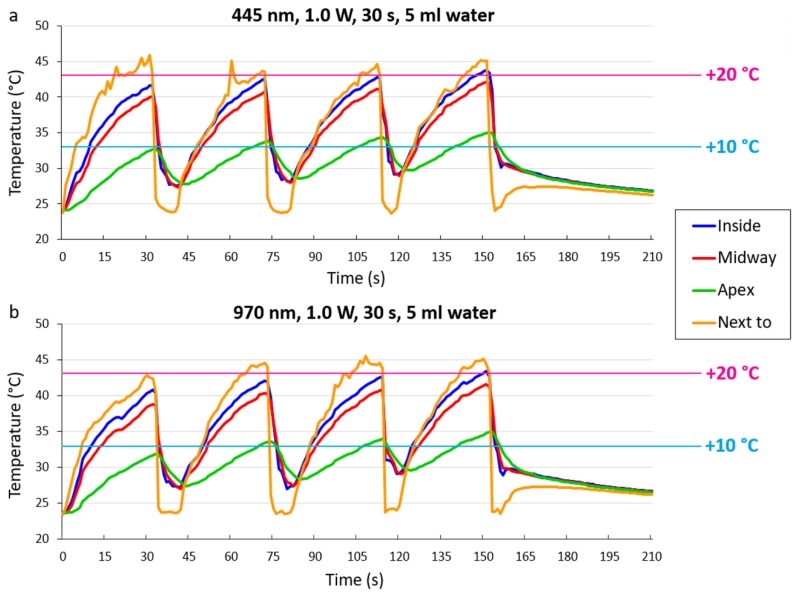
Clinical simulation tests with both wavelengths and thresholds marked in blue for +10 °C and pink for +20 °C.

**Figure 5 materials-12-03934-f005:**
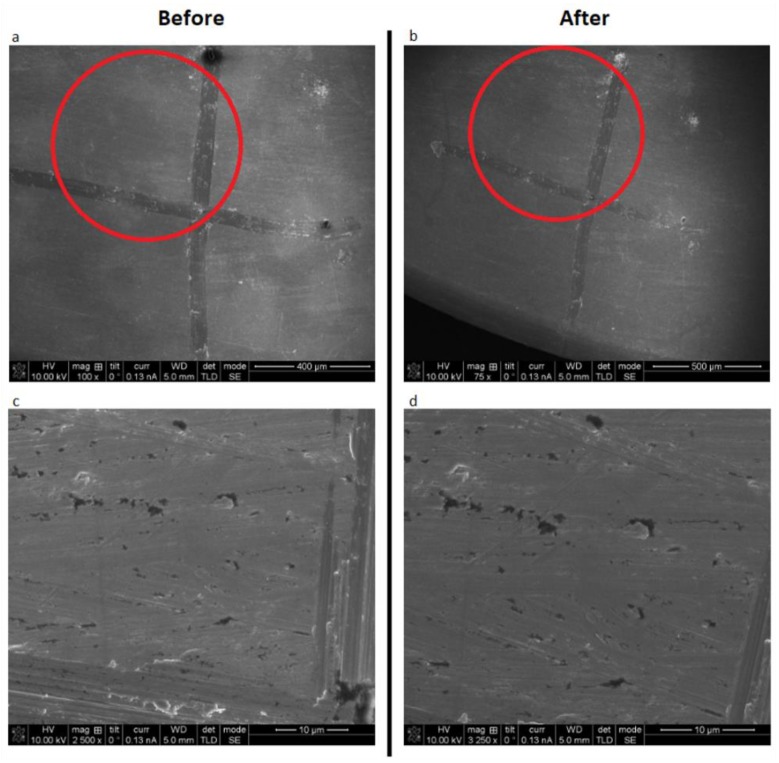
SEM pictures of machined titanium disc before (**a**) and (**c**), as well as after laser irradiation (**b**) and (**d**). Pictures (**a**) and (**b**) have comparable magnification levels, just as (**c**) and (**d**) do.

**Figure 6 materials-12-03934-f006:**
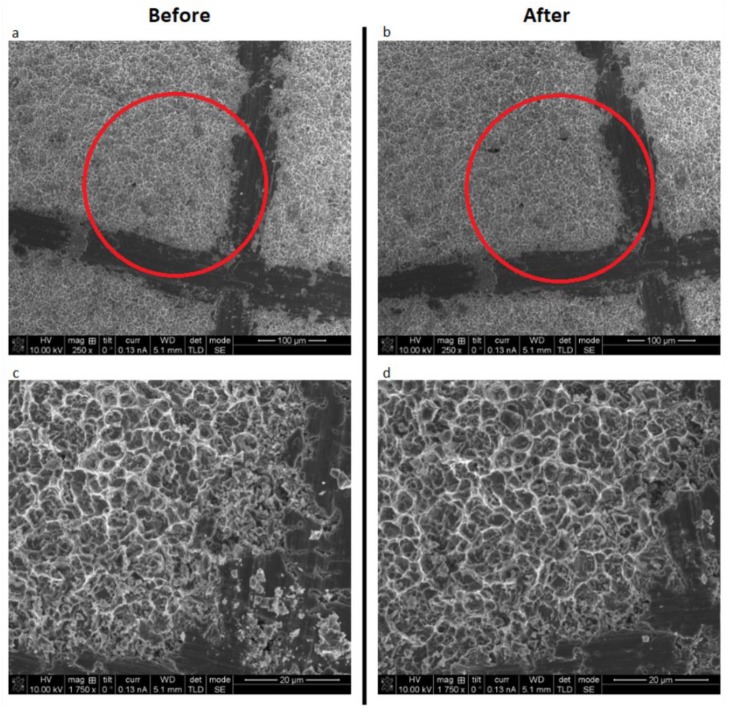
SEM pictures of sandblasted and acid-etched disc before (**a**) and (**c**), as well as after laser irradiation (**b**) and (**d**). Pictures (**a**) and (**b**) have comparable magnification levels, just as (**c**) and (**d**) do.

**Table 1 materials-12-03934-t001:** Average actual power output of the laser device with the 445 nm and 970 nm wavelengths at the displayed power setting of 1.0 W and 2.0 W.

Wavelength	1.0 W	2.0 W
445 nm	1.14 (± 0.02)	2.25 (± 0.03)
970 nm	1.14 (± 0.01)	2.23 (± 0.02)

Values are in average power with standard deviations (SD). W, watt; nm, nanometer.

**Table 2 materials-12-03934-t002:** The average time to reach each of the two thresholds +10 °C and +20 °C, with both wavelengths with 1.0 W power setting continuous wave, at the four different measurement points on the dental implant in the PM model.

Wavelength and Variable	Inside	Midway	Apex	Next to Irradiation Site
445 nm, t to +10 °C	9.00 (± 1.41)	7.00 (± 1.41)	24.00 (± 1.41)	2.00 (± 0.00)
970 nm, t to +10 °C	8.00 (± 0.00)	6.50 (± 0.71)	24.50 (± 0.71)	2.00 (± 0.00)
445 nm, t to +20 °C	27.00 (± 2.83)	20.00 (± 1.41)	75.00 (± 1.41)	5.50 (± 1.41)
970 nm, t to +20 °C	23.50 (± 0.71)	22.00 (± 1.41)	82.50 (± 0.71)	5.50 (± 0.71)

Values are means of three repeated tests with standard deviations (SD) in seconds (s). PM, pig mandible; t, time in seconds; nm, nanometre; °C, degree Celsius; W, watt.

**Table 3 materials-12-03934-t003:** Change in temperature at different time points for both wavelengths with 1.0 W power setting continuous wave, and at the four different measurement points on the dental implant in the PM model.

Wavelength and Variable	Inside	Midway	Apex	Next to Irradiation Site
445 nm, 15 s	14.50 (± 1.56)	17.30 (± 1.13)	6.75 (± 0.64)	28.55 (± 1.20)
970 nm, 15 s	16.00 (± 0.42)	17.15 (± 0.49)	6.70 (± 0.28)	27.90 (± 0.71)
445 nm, 30 s	21.00 (± 0.99)	23.55 (± 0.35)	12.05 (± 0.21)	34.00 (± 0.28)
970 nm, 30 s	22.50 (± 0.42)	22.80 (± 0.71)	11.70 (± 0.28)	33.05 (± 0.49)
445 nm, 60 s	28.15 (± 2.33)	29.05 (± 0.35)	18.00 (± 0.28)	39.70 (± 0.42)
970 nm, 60 s	28.50 (± 0.28)	27.80 (± 0.28)	17.20 (± 0.28)	36.95 (± 0.35)
445 nm, 120 s	34.60 (± 2.12)	34.05 (± 0.07)	23.96 (± 0.07)	44.80 (± 0.42)
970 nm, 120 s	34.70 (± 0.28)	32.75 (± 0.35)	23.05 (± 0.21)	42.15 (± 0.64)

Values are means of three repeated tests with standard deviations (SD) in degrees Celsius (°C). PM, pig mandible; s, seconds; nm, nanometre; W, watt.

**Table 4 materials-12-03934-t004:** The average time to reach each of the two thresholds +10 °C and +20 °C with both wavelengths with 1.0 W power setting continuous wave at the four different measurement points on the dental implant. Comparison of PM and GIC models at 1.0 W with both wavelengths.

Wavelength and Variable	Model	Inside	Midway	Apex	Next to Irradiation Site
445 nm t to +10 °C	PM	9.00 (± 1.41)	7.00 (± 1.41)	24.00 (± 1.41)	2.00 (± 0.00)
	GIC	3.67 (± 0.58)	11.67(± 0.58)	25.00 (± 0.00)	3.00 (± 0.00)
970 nm t to +10 °C	PM	8.00 (± 0.00)	6.50 (± 0.71)	24.50 (± 0.71)	2.00 (± 0.00)
	GIC	4.00 (± 0.00)	12.00 (± 0.00)	25.33 (± 0.58)	3.33 (± 0.58)
445 nm Max Δ temp	PM	34.90 (± 2.12)	34.25 (± 0.07)	24.25 (± 0.07)	45.05 (± 0.35)
	GIC	87.83 (± 0.65)	36.93 (0.25)	27.07 (± 0.15)	61.00 (± 0.36)
970 nm Max Δ temp	PM	34.95 (± 0.35)	32.95 (± 0.35)	23.40 (± 0.28)	42.20 (± 0.57)
	GIC	87.23 (± 0.12)	36.77 (± 0.06)	26.57 (± 0.06)	58.63 (± 0.40)

Values are means of three repeated tests with standard deviations (SD). PM, pig mandible; GIC, glass ionomer cement; t, time in seconds; nm, nanometre; ΔT, change in temperature in degree Celsius.

**Table 5 materials-12-03934-t005:** Comparison of maximum temperatures (°C) between pulsed and continuous wave mode of 445 nm laser.

Settings	Inside	Midway	Apex	Next to Irradiation Site
0.5 W CW	45.97 (± 0.06)	19.30 (± 0.00)	13.90 (± 0.00)	32.00 (± 0.36)
3 W Pulsed 17% Duty Cycle 10 Hz (avg 0.51 W)	51.60 (± 0.10)	28.80 (± 0.00)	13.93 (± 0.06)	37.00 (± 0.20)

Values are means of three repeated tests with standard deviations (SD) in degrees Celsius. W, watt; nm, nanometre; cw, continuous wave; Hz, hertz °C, degree Celsius; avg, average.

**Table 6 materials-12-03934-t006:** Differences in maximum temperature after 2 min of irradiation at 1.0 W, when the fibre tip is in contact with, and at 3 mm distance to, the dental implant: ∆T_max_ = T_max_ (3 mm) − T_max_ (0 mm).

Wavelength	Inside	Midway	Apex	Next to Irradiation Site
445 nm	−10.3	−5.9	−3.5	2.5
970 nm	−5.4	−3.1	−1.8	2.1

Values are in degree Celsius (°C). W, watt; mm, millimetre; T_max_, maximum temperature.

**Table 7 materials-12-03934-t007:** Average temperatures between time points 15 s and 120 s when continuously applying water on the irradiation site for settings 1.0 W and 2.0 W with both wavelengths.

Power	Wavelength	Inside	Midway	Apex	Next to Irradiation Site
1.0 W	445 nm	37.68 (± 1.53)	26.01 (± 0.50)	30.72 (± 0.50)	32.49 (± 2.61)
	970 nm	42.02 (± 4.43)	27.05 (± 0.72)	33.96 (± 2.45)	32.95 (± 3.86)
2.0 W	445 nm	52.45 (± 2.92)	28.93 (± 1.02)	39.96 (± 1.63)	40.86 (± 3.50)
	970 nm	61.89 (± 6.66)	31.04 (± 1.09)	46.00 (± 3.70)	47.76 (± 3.81)

Values are means of three repeated tests with standard deviations (SD) in degrees Celsius (°C). W, watt; s, seconds; nm, nanometre.

**Table 8 materials-12-03934-t008:** Average temperature reduction in percent after 30 s interval of laser irradiation at 1.0 W when applying 5.0 mL or 2.5 mL cooling water.

Wavelength	Amount of Water	Inside	Midway	Apex	Next to Irradiation Site
445 nm	5.0 mL	82.92 (± 1.98)	50.91 (± 7.67)	61.74 (± 7.02)	91.60 (± 2.19)
	2.5 mL	61.72 (± 4.18)	48.07 (± 4.69)	38.62 (± 4.04)	69.44 (± 4.83)
970 nm	5.0 mL	78.56 (± 1.92)	63.74 (± 4.58)	63.10 (± 4.23)	89.76 (± 4.75)
	2.5 mL	63.09 (± 8.01)	49.16 (± 9.20)	42.34 (± 7.13)	70.43 (± 8.49)

Values are mean reductions in percent (%) of three repeated tests with standard deviations (SD). W, watt; ml, millimetre; nm, nanometre.

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
