# Peer review of "Using 445 nm and 970 nm Lasers on Dental Implants—An In Vitro Study on Change in Temperature and Surface Alterations"

_materials, 2019, doi:10.3390/ma12233934_

Round 1
Reviewer 1 Report
The authors are advised to describe and discuss the manufacturer’s recommendation while evaluating the safety of using a 445 nm laser on dental implants.
Does this study recommend the higher peak power with the pulsed mode?
Why the samples were irradiated with the 445 nm wavelength for “4 minutes” to do surface analysis using SEM unlike the other two models
The authors are encouraged to rewrite the conclusion of this study rather than offering clinical usage recommendations.
Author Response
Dear Reviewer,
Thank you for your feedback on our manuscript entitled “Using 445 nm and 970 nm lasers on dental implants - an in vitro study on change in temperature and surface alterations”. Below we list point-by-point the revisions made and our responses to the comments.
Point 1: The authors are advised to describe and discuss the manufacturer’s recommendation while evaluating the safety of using a 445 nm laser on dental implants.
Response 1: The manufacturer does not have any recommended settings for using 445 nm laser on dental implants.
Point 2: Does this study recommend the higher peak power with the pulsed mode?
Response 2: No, we recommend using continuous mode. We compare continuous with high peak power pulsed because they differ more than a high frequency pulsed setting and continuous does. It is of interest to compare different frequencies and peak powers and see how they affect the temperature, but this was not the main focus of this study. It is however an interesting topic for another project.
Point 3: Why the samples were irradiated with the 445 nm wavelength for “4 minutes” to do surface analysis using SEM unlike the other two models
Response 3: We started with 4 minutes and saw no alterations, so we did not perform a 2 minutes test. 4 minutes of continuous irradiation makes it in our opinion distinctly clear that it does not cause alterations.
Point 4: The authors are encouraged to rewrite the conclusion of this study rather than offering clinical usage recommendations.
Response 4: The conclusion has been changed.
Reviewer 2 Report
This paper evaluated the safety of using 445 nm laser on dental implant over 970 nm laser. The following comments should be addressed before the paper can be considered further:
Line 43, the sentence “Surgical lasers represent different types of lasers, depending the mechanism by which the laser light is generated” is not clear and requires further elaborations. Line 44, the sentence “the wavelength of the laser, which in turn defines the absorption in the tissue” is not clear and some explanations are required. A scale bar should be added to Fig 1. Scale bars should be shown in each image to improve image quality.
Author Response
Dear Reviewer,
Thank you for your feedback on our manuscript entitled “Using 445 nm and 970 nm lasers on dental implants - an in vitro study on change in temperature and surface alterations”. Below we list point-by-point the revisions made and our responses to the comments.
Point 1: Line 43, the sentence “Surgical lasers represent different types of lasers, depending the mechanism by which the laser light is generated” is not clear and requires further elaborations. Line 44, the sentence “the wavelength of the laser, which in turn defines the absorption in the tissue” is not clear and some explanations are required.
Response 1: We completely agree that the previous sentence required revision. We have revised it and the sentence after that one on rows 43-48 in the track changes document. Hopefully the changes made makes our point clearer and will improve the flow of that part of the introduction.
Point 2: A scale bar should be added to Fig 1. Scale bars should be shown in each image to improve image quality.
Response 2: Including a scale bar would, in our opinion, not improve the images as their purpose is to give an overview of the set up and not be a detailed scheme. Another aspect of including scale bars are that e.g. figure 1 has depth to it and a scale bar would not be representative. One could still get a sense of the scale since the dimensions of the glass ionomer block and dental implants are stated in the text. Also, the laser handpiece is visible in both figure 1 and 2 to give some indication of size. In figure 3, which unfortunately was missing in the first version, the discs are stated to have a diameter of 10 mm with a thickness of 2 mm.
We have included scale bars in figures 5 and 6.
Reviewer 3 Report
The in vitro study by Malmqvist et al analyzes the use of 445 nm and 970 nm lasers on dental implants with regard to temperature change and surface alterations. The study showed that no surface alterations could be found at both wavelengths and that maximum temperature and time to reach the threshold were comparable.
The authors analyzed different parameters such as power settings, different models (pig and glass ionomer cement block), different modes (continuous wave or pulsed mode), different forms of irrigation, as well as statistic irradiation and radiation with movement. Furthermore, different distances were analyzed at four different measurement points.
The introduction gives a good overview of the topic and has an adequate length.
The materials and methods section describes the procedure clearly. What is unclear is why the pig models were kept frozen until shortly before the procedure? How old were the pigs? What drilling protocol was used for implant insertion (cortical / cancellous). How was the implant position (vertical position of the implant neck)?
What remains unclear to the reader is how many mandibles / glass ionomer blocks were treated? Were all tests performed only on one mandible and one implant? The tables and figures do not include an n= X
Was only Excel used for the statistics?
Discussion
Can the results be transferred in a clinical study? Does blood in patients influence the results?
Author Response
Dear Reviewer,
Thank you for your feedback on our manuscript entitled “Using 445 nm and 970 nm lasers on dental implants - an in vitro study on change in temperature and surface alterations”. Below we list point-by-point the revisions made and our responses to the comments.
Point 1: The materials and methods section describes the procedure clearly. What is unclear is why the pig models were kept frozen until shortly before the procedure?
Response 1: What is considered short is relative, but the pig mandible was thawed for 2h which was enough for it to reach room temperature. Then the implant was heated according to what is described in the article and the temperature was then allowed to settle again. At the start of the test the implant temperature was stable at room temperature at all 4 measuring sites.
A sentence has been added in rows 152-153 to clarify this.
Point 2: How old were the pigs?
Response 2: One pig mandible was used in this study and the pig was a slaughter pig at the age of 6-7 months.
Point 3: What drilling protocol was used for implant insertion (cortical / cancellous). How was the implant position (vertical position of the implant neck)?
Response 3: The implant was placed in an edentulous area with the neck of the implant 2 mm above the bone level. Drilling procedure was performed according to standard protocol for the specific implant (Astra Tech).
We have added more details regarding this in section 2.2.2 (row 144-153).
Point 4: What remains unclear to the reader is how many mandibles / glass ionomer blocks were treated? Were all tests performed only on one mandible and one implant? The tables and figures do not include an n= X
Response 4: We have states that one block and one pig mandible was used. It is stated in the methods section rows 91-94 that most tests were repeated three times.
We have added clarification to the tables that the tests were repeated three times.
Point 5: Was only Excel used for the statistics?
Response 5: Yes, only Excel was used and since the statistical analyses were of descriptive nature, no more advanced statistical programme was needed.
Point 6: Discussion. Can the results be transferred in a clinical study? Does blood in patients influence the results?
Response 6: Yes, the blood flow in the surrounding tissues would likely affect the results in a clinical study. We have mentioned it on rows 300-301.
Reviewer 4 Report
This paper by Malmqvist et al. reports thermal effects by 445 nm laser on pig mandible and glass ionomer substrates as well as titanium discs in comparison with 970 nm laser. The authors first examined temperature changes of the substrates and then observed surface alteration of titanium discs by SEM before and after laser irradiation.
Experimental setup and results look convincing, and the logical flow of the manuscript is reasonable. Although the originality of the study might not be very significant, their observations seem still be worth publishing. I recommend acceptation for publication after the following points are addressed:
Introduction: the authors should extensively revise this section by mentioning previous studies by other researchers dealing with lasers to clarify the significance of this study. Some sentences can be moved from "Discussion" section.
Figure 3 is missing.
Some typos: l.36, unnecessary "in" before "in line with"; l.181, missing space between "445" and "nm"; l.228 and l.232, should be "2.5" not "2,5"; Table 8, missing +- for SD of 970 nm, Midway.
Author Response
Dear Reviewer,
Thank you for your feedback on our manuscript entitled “Using 445 nm and 970 nm lasers on dental implants - an in vitro study on change in temperature and surface alterations”. Below we list point-by-point the revisions made and our responses to the comments.
Point 1: Introduction: the authors should extensively revise this section by mentioning previous studies by other researchers dealing with lasers to clarify the significance of this study. Some sentences can be moved from "Discussion" section.
Response 1: Added a short paragraph rows 64 to 69 in the introduction to highlight the significance of this study.
Point 2: Figure 3 is missing.
Response 2: Added figure 3 and its figure caption after its paragraph.
Point 3: Some typos: l.36, unnecessary "in" before "in line with"; l.181, missing space between "445" and "nm"; l.228 and l.232, should be "2.5" not "2,5"; Table 8, missing +- for SD of 970 nm, Midway.
Response 3: We have corrected them in the new version of the manuscript.
Round 2
Reviewer 2 Report
The authors have addressed the comments and i recommend acceptance of the manuscript.
Reviewer 4 Report
I think the manuscript has been properly revised.